# Vaspin in Serum and Urine of Post-Partum Women with Excessive Gestational Weight Gain

**DOI:** 10.3390/medicina55030076

**Published:** 2019-03-23

**Authors:** Marcin Trojnar, Jolanta Patro-Małysza, Żaneta Kimber-Trojnar, Monika Czuba, Jerzy Mosiewicz, Bożena Leszczyńska-Gorzelak

**Affiliations:** 1Chair and Department of Internal Medicine, Medical University of Lublin, 20-081 Lublin, Poland; marcin.trojnar@umlub.pl (M.T.); jerzy.mosiewicz@umlub.pl (J.M.); 2Chair and Department of Obstetrics and Perinatology, Medical University of Lublin, 20-090 Lublin, Poland; jolapatro@wp.pl (J.P.-M.); monikaczuba77@o2.pl (M.C.); b.leszczynska@umlub.pl (B.L.-G.)

**Keywords:** adipokines, vaspin, leptin, ghrelin, fatty acid-binding protein 4, excessive gestational weight gain, bioelectrical impedance analysis

## Abstract

*Background and objectives*: Data concerning vaspin in obstetric aspects are limited and conflicting. The aim of the study was to evaluate vaspin concentrations in the serum and urine of women with excessive gestational weight gain (EGWG) in the early post-partum period (i.e., 48 h after delivery), when placental function no longer influences the results. *Materials and Methods*: The study subjects were divided into two groups of 28 healthy controls and 38 mothers with EGWG. Maternal body composition and hydration status were evaluated by the bioelectrical impedance analysis (BIA) method. Concentrations of vaspin, fatty acid-binding protein 4 (FABP4), leptin, and ghrelin were determined via enzyme-linked immunosorbent assay (ELISA). *Results*: Serum vaspin levels were lower in the EGWG group, whereas no significant differences were noted between the groups, with regard to the urine vaspin concentrations. In both studied groups, the serum vaspin concentrations correlated positively with the urine FABP4 levels and negatively with gestational weight gain, body mass index gain in the period from pre-pregnancy to 48 h after delivery (ΔBMI), and fat tissue index (FTI). In the multiple linear regression models, the serum vaspin concentrations were positively dependent on the serum FABP4 levels, as well as negatively dependent on triglycerides, FTI, and ΔBMI. *Conclusions*: Our study revealed that the EGWG mothers were characterized by significantly lower serum vaspin concentrations in the early post-partum period compared with the subjects that had appropriate gestational weight gain. Our observation supports previous hypotheses that vaspin might be used as a marker of lipid metabolism in pregnancy and maternal adipose tissue. Considering the fact that FABP4 is widely referred to as a pro-inflammatory adipokine, further research on the protective role of vaspin seems crucial, especially in the context of its relationship to FABP4.

## 1. Introduction

Human pregnancy is characterized by a gradual increase in insulin resistance from mid-pregnancy until delivery [1]. The release of excess adipokines seems to be implicated in maternal metabolism during pregnancy [2,3]. Insulin sensitivity is typically restored to pre-pregnancy levels in the postpartum period.

Vaspin, also known as serpin A12, is predominantly secreted from visceral adipose tissue. This novel adipokine might also be detected in the stomach, liver, pancreas, hypothalamus, serum, saliva, gingival fluid, and cerebrospinal fluid [4,5]. Vaspin serum concentrations have been associated with glucolipid metabolism [6,7,8,9]. Previous studies have indicated that vaspin acts as an insulin sensitizer with anti-inflammatory effects and might serve as a compensatory mechanism in the pathogenesis of visceral obesity, type 2 diabetes mellitus (T2DM), and atherosclerosis [6,10,11,12,13,14].

Data concerning vaspin in obstetric aspects are limited to pregnant patients with gestational diabetes mellitus (GDM) [1,2,3,6,10,15,16,17,18,19,20,21], preeclampsia [1,21], intrauterine growth restriction [22], and hyperemesis gravidarum [23], as well as to large-for-gestational-age infants [24]. However, no consensus exists regarding vaspin levels in pregnancies complicated by GDM [16]. Some previously published studies detected a significant increase in serum vaspin concentrations in GDM. A few of them found the opposite outcome, whereas the rest of the studies found no association of vaspin levels with the disease [16].

The abovementioned conflicting results seem to deal with varied trial protocols that have been implemented, different inclusion criteria, and a non-homogenous population of pregnant women. It is worth noting that subjects did not present similar body mass indices (BMI), pregnancy stages, or hypoglycemic treatment strategies, with respect to diet or insulin therapy. [15].

It should also be emphasized that circulating vaspin in pregnant females originates from other tissues and organs, such as the placenta [18,25].

Taking into consideration these aforementioned ambiguous and, to some extent, controversial observations, the methodology of our current study was based on a selection of puerperal mothers with normal and excessive gestational weight gain (EGWG). We hypothesized that circulating vaspin concentrations would probably be impaired in the subgroup with abnormally high weight increase in pregnancy. We deliberately decided to concentrate on the puerperium, when placental function no longer influences the results. Having considered the previously reported role of vaspin in weight gain and reduction [26,27,28,29], we analyzed associations between vaspin and additional variables, such as BMI values, ghrelin, leptin, and bioelectrical impedance analysis. We hypothesized that due to the vaspin molecular weight of approximately 45.2 kDa, the presence of this adipokine in urine is highly probable. As far as we know, to date, there is no study investigating vaspin concentrations in human urine. The aim of this study was also to assess whether the serum and urine vaspin levels correlated with selected maternal parameters.

## 2. Materials and Methods

Sixty-six Caucasian females in a singleton term pregnancy (after 37 weeks of gestation), who delivered at the Chair and Department of Obstetrics and Perinatology, at the Medical University of Lublin were enrolled in the study. The data were collected from March 2016 to February 2017. The enrolled subjects were on vitamin–iron supplementation only, during the whole pregnancy. They presented normal pre-pregnancy BMI and showed no carbohydrate metabolism disorders, as documented in three normal results of the two-hour, 75 g oral glucose tolerance test performed at 24–28 weeks of gestation [30,31]. Two groups were selected based on the gestational weight gain:Healthy controls—28 pregnant women with normal gestational weight gain;Patients with EGWG—38 pregnant subjects with excessive gestational weight gain.

The exclusion criteria included: Multiple pregnancy, chronic diseases, current urinary infections, abnormal laboratory findings (e.g., complete blood count, urine test, creatinine, glomerular filtration rate (GFR), metabolic disorders, including polycystic ovarian syndrome (except for improper gestational weight gain for the EGWG group), mental illnesses, cancer, liver disfunction, cardiovascular diseases, known fetal malformation, premature membrane rupture, intrauterine growth retardation, the presence of metallic prostheses, and pacemakers or cardioverter-defibrillators. We had to exclude all pregnant women with gestational hypertension, which represents a relatively common complication in patients with increased BMI in the third trimester of pregnancy.

Anthropometric measurements and sampling were performed after a 6 h fasting in the early post-partum period (i.e., 48 h after delivery). Maternal body composition and hydration status were evaluated by means of the bioelectrical impedance analysis (BIA) method, with the use of a body composition monitor (BCM) (Fresenius Medical Care). Calculation of pre-pregnancy BMI values was based on body weight measured at the first prenatal visit, occurring in the first trimester (before 10 weeks of gestation). We defined total gestational weight gain as the difference between the mother’s weight at delivery and her pre-pregnancy weight. We calculated gestational BMI gain as well. The maternal serum levels of albumin, fasting blood glucose, hemoglobin A1c, and lipid profile were measured by a certified laboratory. The maternal serum and urine samples were taken after delivery, taking into account a 6 h fasting period. After centrifugation, all the collected maternal serum and urine samples were stored at −80 °C. The concentrations of vaspin (MyBioSource.com, San Diego, CA, USA), fatty acid-binding protein 4 (FABP4; R and D Systems, Inc., Minneapolis, MN, USA), leptin (R and D Systems, Inc., Minneapolis, MN, USA), and ghrelin (Wuhan EIAab Science Co., Wuhan, China) in these materials were determined using commercially available kits in compliance with the manufacturer’s instructions via traditional enzyme-linked immunosorbent assay (ELISA). The minimum detectable dose of human leptin is typically less than 7.8 pg/mL. Due to the urine leptin levels obtained in the majority of the patients, which were below the threshold of the sensitivity of the ELISA test, the “urine leptin” parameter is not present in the results. The survey was performed in duplicates for each patient.

All the patients were informed about the study protocol and a detailed written consent was obtained from each patient who agreed to participate in the study. The study protocol received approval from the Bioethics Committee of the Medical University of Lublin (no. KE-0254/221/2015 (approved on 25th June 2015) and no. KE-0254/348/2016 (approved on 15 December 2016)).

All values were reported as the median (interquartile range 25–75%) or numbers and percentages. Differences between groups were tested for significance using the Mann–Whitney U test. Spearman’s coefficient test was used for the correlation analyses. The multiple linear regression model was used to adjust the covariates and examine the associations between the serum vaspin levels and the selected maternal biophysical and biochemical parameters. All analyses were performed using Statistical Package for the Social Sciences software (version 19; SPSS Inc., Chicago, IL, USA). A *p*-value of <0.05 was considered statistically significant.

## 3. Results

Compared with the healthy study participants, the EGWG mothers presented significantly higher values of body mass index (BMI) at the time of and after delivery, as well as gestational weight and BMI gains, and BMI gain in the period from pre-pregnancy to 48 h after delivery (ΔBMI), but they were of comparable age, had pre-pregnancy BMIs, and BMI loss after delivery. Some of the results regarding the characteristics of the study subjects were previously presented by us [32]. The EGWG women were characterized by increased levels of hemoglobin A1c, triglycerides, total body water (including extracellular (ECW), as well as intracellular (ICW) water), body cell mass, and indexes of fat (FTI) and lean (LTI) tissues, as well as lower concentrations of high-density lipoprotein cholesterol (HDL). While lower serum vaspin concentrations were observed in the EGWG mothers, no significant differences were noted between the groups with regard to other analyzed parameters, including urine vaspin levels (Table 1, Figure 1).

A positive correlation was found between the serum vaspin concentrations and the urine FABP4 levels in both of the studied groups. The serum vaspin levels also correlated positively with BMI loss after delivery, but only in the EGWG mothers.

Negative correlations between the serum vaspin concentrations and gestational weight gain, ΔBMI, and FTI were observed in the control and EGWG groups. The serum vaspin level was negatively associated with gestational BMI gain, albumin, hemoglobin A1c, triglycerides, serum ghrelin and leptin, and urine vaspin levels only in the healthy mothers. Inverse correlations were also found between the serum vaspin and BMIs, at the time of and after delivery, and ECW in the EGWG group (Table 2).

Positive correlations were present between the urine vaspin and total cholesterol as well as between the LDL and triglycerides levels in both studied groups. The urine vaspin concentrations also correlated positively with gestational BMI gains, the serum albumin, hemoglobin A1c and ghrelin levels, and FTIs in the control subjects. Furthermore, direct correlations were noted between the urine vaspin and pre-pregnancy BMI values and the urine ghrelin levels in the EGWG group.

The urine vaspin concentrations correlated negatively with the serum vaspin and ECW in the control subjects, as well as with the total body water, ECW, and ICW in the EGWG mothers (Table 2).

In the multiple linear regression models, after adjustment for the serum FABP4 levels, the urine vaspin levels, HDL, low-density lipoprotein cholesterol (LDL), triglycerides, hemoglobin A1c, FTI, and ΔBMI, we noted that the serum vaspin concentrations were positively dependent on the serum FABP4 levels, while being negatively dependent on triglycerides, FTI, and ΔBMI (Table 3).

Adjusted for serum FABP4 levels, the urine vaspin levels, high-density lipoprotein cholesterol, low-density lipoprotein cholesterol, triglycerides, hemoglobin A1c, fat tissue index and ΔBMI. Unstandardized β coefficients with 95% confidence interval and B linear regression coefficients are shown. Statistically significant values are given in bold type. ΔBMI—body mass index gain in the period from pre-pregnancy to 48 h after delivery; FABP4—fatty acid-binding protein 4.

## 4. Discussion

Our study proved that the EGWG mothers were characterized by significantly lower serum vaspin concentrations in the early post-partum period (i.e., 48 h after delivery) compared with the subjects who had appropriate gestational weight gain. We cannot relate our results to any previous results since there are no data regarding relationship between vaspin and abnormal weight gain during pregnancy available. Nevertheless, due to the fact that in many ways, pregnancies of women with EGWG resemble pregnancies complicated by GDM, we tried to discuss our findings in light of the observations made regarding puerperal GDM mothers [2,18]. It was found that vaspin concentrations were significantly reduced after placental delivery, which backs the hypothesis that the placenta does influence the circulating levels of this adipokine during pregnancy [2]. There is ample evidence to suggest that expression of vaspin occurs in cytotrophoblasts and syncytiotrophoblasts in early pregnancy, whereas in the third trimester, this adipokine is only detectable in syncytiotrophoblasts [25].

Huo et al. [18] evaluated vaspin plasma levels and its placental expression in pregnant GDM patients and women with uncomplicated pregnancies. The authors observed that the circulating vaspin levels were lower in the GDM group compared with the control subjects. The serum vaspin levels significantly decreased three days after cesarean section, but only in the GDM mothers [18]. However, there were no statistically significant differences between placental mRNA vaspin levels and protein expression in both groups [18].

The comparable results were obtained by Gkiomisi et al. [2], who measured vaspin levels in women with and without GDM at the second and third trimesters of pregnancy, as well as after delivery. The serum vaspin concentrations were significantly decreased in the postpartum period in comparison with the second and third trimesters in both studied groups; however, the serum vaspin levels were lower in the control subjects than in the GDM group [2].

During normal pregnancy, vaspin seems to mainly be expressed in both the adipose tissue and the placenta. Circulating vaspin concentrations increase throughout gestation, reaching the highest level at the end of the third trimester [25]. At 24–30 weeks of gestation, healthy pregnant women have significantly lower vaspin serum levels than non-pregnant controls [33]. However, vaspin levels slowly decrease from the second trimester until delivery in patients with GDM.

It is still not clear why the concentration of vaspin decreases in the group of GDM women at term. There is no consensus regarding any mechanism in which vaspin alters glucose metabolism or insulin sensitivity, either. Blüher [26] proposed an interaction with a protease. In different in vitro research, human kallikrein 7 (hK7) was found to be the first target protease of vaspin, which confirmed strong inhibitory properties of vaspin. HK7 is known to be engaged in the process of insulin degradation, which seems in line with the observed increased insulin concentration of isolated pancreatic islets treated with vaspin [33]. An interesting study by Nakatsuka et al. [34] confirmed that vaspin significantly improved insulin sensitivity in vivo, in an experiment carried out on transgenic mice. The authors concluded that vaspin secreted by the adipose visceral tissue targeted hepatocytes by interacting with 78 kDa glucose-regulated protein (GRP78) molecules. In the postulated mechanism of action, the mentioned adipokine is believed to modulate the endoplasmic reticulum stress responses, acting as a ligand for plasma membrane-associated GRP78. The observed beneficial metabolic outcome was related to decreased gluconeogenesis and the limited synthesis of fatty acids. A detailed assessment of the metabolic impact of vaspin was performed in an animal model of abdominal obesity with T2DM. The authors proved that expression of this adipokine in rat adipocytes was high when obesity, animal body weight, and insulin levels peaked in the experiment. The tissue expression of vaspin decreased as diabetes worsened, whereas administration of vaspin to obese mice fed with high-fat high-sucrose chow improved glucose tolerance and insulin sensitivity [11].

In our study, an increase in the concentration of triglycerides by 1 mg/dL was associated with a simultaneous decrease in the serum vaspin level by 5.4 pg/mL. Our observation supports the hypothesis of Giomisi et al. [35], that vaspin might be used as a surrogate marker of lipid metabolism in pregnancy. The authors found a negative correlation between vaspin and lipid parameters (i.e., total cholesterol, triglycerides, and LDL) in pregnant patients. Accordingly, a previous report [21] showed that circulating vaspin levels were significantly correlated with parameters of adiposity, including BMI values, the homeostasis model assessment of insulin resistance (HOMA-IR), and lipid profile (i.e., total cholesterol and triglycerides). Similarly, Engin-Ustun et al. [23] observed that the serum vaspin concentrations inversely correlated with total cholesterol, triglyceride, LDL, and very low-density lipoprotein cholesterol in pregnant patients with hyperemesis gravidarum.

Taking into account associations between vaspin and body weight parameters, we found that the serum vaspin levels correlated negatively with gestational weight gain and ΔBMI in both studied groups. Multiple linear analyses revealed that each 1 kg/m^2^ of BMI gain in the period from pre-pregnancy to 48 h after delivery was linked to a decrease in the maternal serum vaspin level by 612.5 pg/mL.

Some authors have also suggested a link between weight loss and vaspin concentrations, however the obtained data is not conclusive [26,27,28,29]. It is worthwhile, however, to mention a study in which Jian et al. [36] found that vaspin levels correlated with BMI and waist–hip ratio in T2DM. The cited authors observed that lower levels of vaspin were a risk factor for the development of T2DM [36]. It is worth highlighting that EGWG mothers in our study (characterized by greater increase in gestational weight gain and BMI, which represented the inclusion criteria for this subgroup) were found to have lower serum vaspin levels and larger values of ΔBMI. The change in BMI, as defined by the difference between its value in the-pre-pregnancy period and that at (or up to) 48 hours after delivery, represents a very important predictor of the future body composition change in mothers. A well-documented meta-analysis focusing on more than 69,000 women revealed that two decades after delivery, abnormally high body mass affected those females who had been diagnosed with excessive weight gain in pregnancy [37]. As a matter of fact, the extent of the weight that is retained following delivery differs considerably between various women. It is obviously related to the extent of weight gain in pregnancy, however it should be stressed that EGWG women are likely to retain more weight [37,38].

Our study also revealed certain interesting associations between vaspin and other adipokines. We observed a positive correlation between the serum vaspin and urine FABP4 levels in both studied groups. Moreover, in the group of mothers with appropriate gestational weight gains, the serum vaspin concentration was negatively associated with serum ghrelin and leptin levels, whereas the urine vaspin concentration was positively linked to serum ghrelin levels. Interdependence between vaspin and other adipokines and cytokines is scarcely presented in literature. However, Giomisi et al. [ 35] found a positive correlation between vaspin and adiponectin in pregnant as well non-pregnant women. The authors also proposed a protective role of vaspin, in case of GDM.

Furthermore, it is interesting to point out the relationship between vaspin and FABP4. Apart from our previously mentioned results from Spearman’s coefficient test, we performed multiple linear analyses that revealed the highly statistically significant dependence of the serum FABP4 concentrations on the serum vaspin levels. Due to the fact that FABP4 seems to be the most promising predictor of pregnancy metabolic complications, the association between vaspin and FABP4 should be highlighted. Liu et al. [39] observed that vaspin was able to promote the expression of FABP4 by increasing the expression of transcription factors, such as peroxisome proliferator-activated receptor γ (PPAR-γ), CCAAT/enhancer-binding protein β (C/EBPB), and α (C/EBPA) in a dose-dependent manner. These effects can promote the differentiation of 3T3-L1 preadipocytes. There is some emerging evidence to prove that vaspin possesses anti-inflammatory potential. It has been demonstrated that this adipokine suppresses cytokine-induced inflammation in 3T3-L1 adipocytes via inhibition of the nuclear factor kappa-light-chain-enhancer of activated B cells (NFkB) pathway. Chronic inflammation and excessive macrophage infiltration are typical features of adipose tissue dysfunction in obesity. In the aforementioned experiment, vaspin was found to attenuate an interleukin 1β (IL-1β)-induced pro-inflammatory cytokine response via inhibition of the NFkB signaling pathway in isolated adipocytes [40]. Nevertheless, these observations may be related to relationships that are not yet fully understood between vaspin and other adipokines, including the abovementioned FABP4. In light of the recent study by Wang et al. [41], it seems to be very interesting to investigate associations between vaspin and fibroblast growth factor-21 (FGF-21) in future research. The authors demonstrated that FGF-21 alleviates inflammation and that the anti-inflammatory effects of FGF-21 are mediated by the fibroblast growth factor receptor substrate 2/extracellular signal-regulated kinase 1/2 (FRS2/ERK1/2) signaling pathway in 3T3-L1 preadipocytes. Their study confirmed a protective effect of FGF-21 on obesity-related diseases [41]. The PI3K/AKT signaling pathway may be involved in this process. Additionally, the administration of vaspin led to significantly smaller adipocytes, reduced expression of IL-6, and increased expression of GLUT4, suggesting that vaspin could potentially be used to increase insulin sensitivity and inhibit obesity.

Considering the fact that FABP4 is widely referred to as a pro-inflammatory adipokine, further research on the protective role of vaspin seems crucial and inevitable, especially in the context of its relationship to FABP4. Our previous findings revealed that the serum and urine FABP4 concentrations correlated negatively in the healthy puerperal mothers [42]. In the present study, comparable results were obtained for vaspin because a negative correlation between its levels in the serum and urine was found only for mothers with appropriate gestational weight gain. Unfortunately, although the study exclusion criteria comprised abnormal GFR and urine tests, particularly proteinuria, the methodology of our study did not include more specialist techniques, such as electrophoresis and flow cytometry of urine proteins. It should also be emphasized that future evaluation of adipokines in urine should be adjusted according to urine creatinine concentrations or daily excretion rate.

The BIA method was used to assess the composition of maternal body and hydration status. This technique represents a non-invasive, reliable, and fast clinical approach, which is well tolerated and widely accepted by patients [43,44,45]. This standardized method has often been used by many researchers [44,46]. It seems that BIA has a better prognostic potential for gestational and post-partum outcomes than BMI [47]. BMI, along with gestational weight gain, indeed informs of the nutritional status of the pregnancy, however theses parameters lack information regarding the distribution of fat [47]. Body fat composition, on the other hand, can be assessed in detail, using BIA. Nowadays, fat and free-fat masses are known to be more accurate predictors than BMI of maternal nutritional status [47]. This study revealed that FTI, which is defined as the adipose tissue mass divided by the square of the body height and is expressed in units of kg/m^2^, was negatively associated with the serum vaspin levels in healthy and EGWG groups. FTI was also positively associated with urine vaspin concentrations, but only in the control subjects. The multiple linear regression model showed that the serum vaspin levels were negatively dependent on FTI.

It has been estimated that EGWG subjects present a threefold increase in abdominal obesity incidence, when evaluated eight years after delivery, compared to those females with adequate pregnancy-related body mass changes [48]. The resulting abdominal obesity plays a role in the progression of carbohydrate metabolism disorders, including impaired glucose tolerance, decreased insulin sensitivity, and T2DM, as well as increased cardiovascular risk [38,49].

Gkiomisi et al. [2] have suggested that vaspin could be a marker of maternal adipose tissue. Many researchers have shown that both serum vaspin levels [50,51] and vaspin expression in adipose tissue [12,13] are closely related to metabolic parameters and insulin resistance [10]. Furthermore, the accommodation mechanism of vaspin gene expression appears to be fat depot-specific. Vaspin mRNA expression was detectable in 23% of the visceral adipose tissue (VAT) samples, but only 15% of the subcutaneous adipose tissue (SAT) samples [12]. In addition, VAT was supposed to be a major contributor to metabolic risk and insulin resistance [52,53].

Our study methodology relied on an accurate selection of the study subjects. We decided to choose EGWG and not pre-pregnant obese women. In the case of pre-pregnancy obesity, analysis of the results would have had to take into consideration the influence of modulators, such as dyslipidemia, hypertension, insulin resistance, pre-pregnancy treatment of obesity, and disorders of the carbohydrate–lipid balance. Because we chose the EGWG group, we were able to reduce the number of interfering and confounding factors in the analysis of the study results. The study groups were formed on the basis of the women’s similar age, normal pre-pregnancy BMIs, and term pregnancies. Selected women were not diagnosed with any chronic or gestational diseases and received only vitamins throughout their pregnancies. On the other hand, this study has, in fact, an important limitation, which is a relatively small sample protocol. Our results require further verification.

## 5. Conclusions

Our study revealed that the EGWG mothers were characterized by significantly lower serum vaspin concentrations in the early post-partum period (i.e., 48 h after delivery), compared with the subjects who had appropriate gestational weight gain.

Our observation supports previous hypotheses that vaspin might be used as a surrogate marker of lipid metabolism in pregnancy, as well as a biomarker of maternal adipose tissue.

Considering the fact that FABP4 is widely referred to as a pro-inflammatory adipokine, further research on the protective role of vaspin seems crucial and inevitable, especially in the context of its relationship to FABP4.

## Figures and Tables

**Figure 1 medicina-55-00076-f001:**
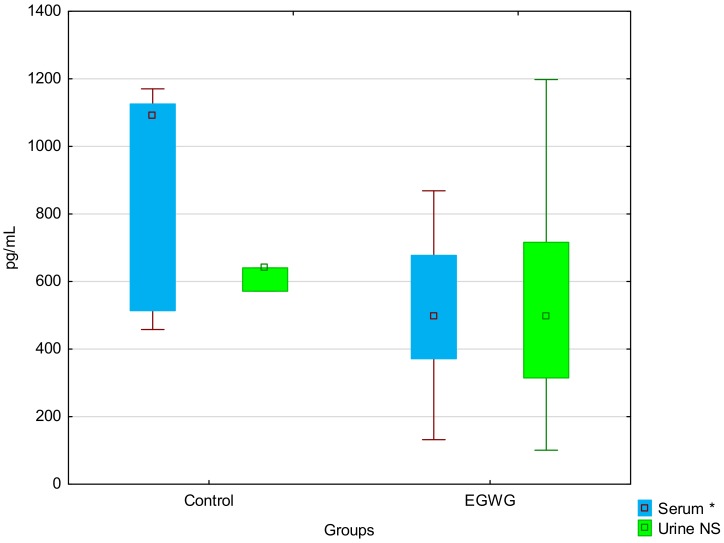
Vaspin levels in maternal serum and urine of the control and excessive gestational weight gain (EGWG) groups, * *p* < 0.05. Vaspin levels in maternal serum and urine of the control and excessive gestational weight gain (EGWG) groups; * *p* < 0.05;—median; boxes—25th to 75th percentiles; whiskers—non-outlier range.

**Table 1 medicina-55-00076-t001:** Comparison of characteristics of the study subjects.

Variables	Control Group (*n* = 28)	EGWG Group (*n* = 38)	*p*
age, years	29 (24–38)	29 (28–32)	NS
pre-pregnancy BMI, kg/m^2^	20.3 (19.5–24.4)	23.2 (21.6–24.09)	NS
**Day of Delivery**			
fasting blood glucose, mg/dL	83.5 (73–91)	80.55 (78–86)	NS
gestational weight gain, kg	15 (11.5–15.6)	23.9 (21–26)	**<0.001**
gestational BMI gain, kg/m^2^	5.4 (3.0–5.6)	8.4 (7.07–9.4)	**<0.001**
BMI at delivery, kg/m^2^	26.3 (24.2–29.1)	31.3 (29.7–32.05)	**<0.001**
**Second Day of Post-Partum Period**			
BMI after delivery, kg/m^2^	22 (21–23.9)	28.6 (26.2–29.7)	**<0.001**
BMI loss after delivery, kg/m^2^	2.5 (2.08–4.16)	2.75 (2–3.2)	NS
ΔBMI, kg/m^2^	0.85 (0.23–2.9)	5.24 (4.24–4.85)	**<0.001**
cesarean delivery, %	14	26	NS
albumin, g/dL	3.68 (3.43–3.73)	3.55 (3.41–3.81)	NS
hemoglobin A1c, %	5.3 (4.6–5.4)	5.5 (5.0–5.5)	**<0.05**
total cholesterol, mg/dL	249 (188–287)	225 (197–249)	NS
HDL, mg/dL	78 (75–82)	71 (59–79)	**<0.05**
LDL, mg/dL	129 (93–152)	106 (87–128)	NS
triglycerides, mg/dL	177 (150–254)	204 (178–258)	**<0.05**
serum FABP4, ng/mL	10.99 (10.63–11.56)	11.97 (10.2–15.48)	NS
serum ghrelin, ng/mL	0.933 (0.646–1.115)	1.187 (0.343–2.433)	NS
serum leptin, ng/mL	10.43 (6.04–14.9)	14.87 (12.6–47.6)	NS
urine FABP4, ng/mL	0.04 (0.03–0.1)	0.03 (0.01–0.08)	NS
urine ghrelin, ng/mL	0.102 (0.096–0.288)	0.116 (0.044–0.303)	NS
total body water, L	30.1 (25.2–35)	34.5 (33.6–41.1)	**<0.001**
extracellular water, L	14.9 (13–15.7)	16.6 (15.5–20)	**<0.001**
intracellular water, L	15.7 (13.5–17.8)	18.5 (17.5–20)	**<0.001**
body cell mass, kg	15 (12.8–20.1)	19.7 (16.9–21.1)	**<0.01**
lean tissue index, kg/m^2^	10.1 (9.4–13.1)	12.9 (11.2–13.9)	**<0.01**
fat tissue index, kg/m^2^	10.1 (9.1–13.8)	14.7 (13.2–17.2)	**<0.001**

The results are shown as the median (interquartile range: 25–75%). Statistically significant values are given in bold. BMI—body mass index; ΔBMI—body mass index gain in the period from pre-pregnancy to 48 h after delivery; EGWG—excessive gestational weight gain; FABP4—fatty acid-binding protein 4; HDL—high-density lipoprotein cholesterol; LDL—low-density lipoprotein cholesterol; NS—not significant.

**Table 2 medicina-55-00076-t002:** Correlations between the maternal serum and urine vaspin levels and the selected parameters.

	Control Group				EGWG Group			
	Serum Vaspin		Urine Vaspin		Serum Vaspin		Urine Vaspin	
pre-pregnancy BMI	0.2	0.398	–0.103	0.667	−0.357	0.074	0.41	**<0.05**
gestational weight gain	–0.8	**<0.001**	0.41	0.072	−0.524	**<0.01**	–0.023	0.909
gestational BMI gain	–0.9	**<0.001**	0.564	**<0.01**	–0.219	0.281	–0.281	0.165
BMI at delivery	–0.2	0.398	–0.103	0.667	–0.462	**<0.05**	–0.179	0.382
**2nd Day of Post-Partum Period**								
BMI after delivery	–0.2	0.398	–0.103	0.667	–0.619	**<0.001**	–0.187	0.359
BMI loss after delivery	–0.3	0.199	0.051	0.829	0.446	**<0.05**	0.074	0.718
ΔBMI	–0.8	**<0.001**	0.41	0.072	–0.572	**<0.01**	–0.342	0.088
albumin	–0.7	**<0.001**	0.462	**<0.05**	0.058	0.779	0.309	0.125
hemoglobin A1c	–0.564	**<0.01**	0.553	**<0.05**	0.123	0.565	–0.025	0.909
total cholesterol	–0.3	0.199	0.564	**<0.01**	0.022	0.915	0.481	**<0.05**
HDL	–0.1	0.675	0.359	0.119	–0.083	0.688	0.225	0.269
LDL	–0.3	0.199	0.564	**<0.01**	0.077	0.708	0.493	**<0.05**
triglycerides	–0.564	**<0.01**	0.605	**<0.01**	0.187	0.361	0.525	**<0.01**
serum vaspin			–0.821	**<0.001**			–0.352	0.078
serum FABP4	–0.4	0.125	0.316	0.233	0.429	0.165	0.486	0.109
serum ghrelin	–0.8	**<0.001**	0.949	**<0.001**	–0.029	0.929	–0.314	0.319
serum leptin	–0.6	**<0.05**	0.316	0.233	0.486	0.109	–0.143	0.658
urine vaspin	–0.821	**<0.001**			–0.352	0.078		
urine FABP4	0.6	**<0.05**	–0.316	0.233	0.6	**<0.05**	–0.257	0.419
urine ghrelin	0.0000001	1.0	0.316	0.233	0.086	0.791	0.714	**<0.01**
total body water	0.0000001	1.0	–0.41	0.072	–0.298	0.139	–0.47	**<0.05**
extracellular water	0.3	0.199	–0.667	**<0.01**	–0.498	**<0.01**	–0.452	**<0.05**
intracellular water	–0.2	0.398	–0.308	0.187	–0.022	0.915	–0.405	**<0.05**
body cell mass	–0.2	0.398	–0.308	0.187	0.272	0.178	–0.204	0.318
lean tissue index	0.0000001	1.0	–0.41	0.072	0.121	0.555	0.001	0.995
fat tissue index	–0.5	**<0.05**	0.462	**<0.05**	–0.599	**<0.01**	0.025	0.904

Statistically significant values are given in the bold type. BMI—body mass index; ΔBMI—BMI gain in the period from pre-pregnancy to 48 h after delivery; EGWG—excessive gestational weight gain; FABP4—fatty acid-binding protein 4; HDL—high-density lipoprotein cholesterol; LDL—low-density lipoprotein cholesterol.

**Table 3 medicina-55-00076-t003:** Multiple linear regression analyses for serum vaspin levels.

Variables	B	β	95% CI	*p*
serum FABP4	232	1.7	1.1-2.3	**<0.001**
triglycerides	−5.4	−0.6	−1–(−0.24)	**<0.01**
fat tissue index	−929	−3.6	−5–(−2.2)	**<0.001**
ΔBMI	−612.5	−3.2	−4.5–(−2)	**<0.001**

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
