# Peer review of "Vaspin in Serum and Urine of Post-Partum Women with Excessive Gestational Weight Gain"

_medicina, 2019, doi:10.3390/medicina55030076_

Round 1

Reviewer 1 Report

Trojnar et al. report a small pilot study that addressed vaspin concentrations in serum and urine of women with excessive gestational weight gain (EGWG). The authors found significantly lower vaspin levels in EGWG women and report interesting data that suggests a relationship of circulating vaspin and FABP4. Unfortunately, the authors fall short of discussing their data with respect to the recent insight gained on vaspin function in vitro and in vivo. Many related and highly relevant and interesting papers with respect to the authors findings are not discussed.

Instead, citation of review literature and in general papers only marginally related to the topic is excessive. The authors should focus on the relevant primary literature. This clearly decreases the enthusiasm for this manuscript and does not help the interested reader.

This is especially important, when introducing and discussing vaspin function to interpret the authors findings. Yet, findings of important papers are not discussed (e.g. Nakatsuka et al. 2012 Diabetes-> vaspin KO and tg mice, identification of GRP78 as cell surface receptor; Heiker et al. 2013 Cell Mol Life Sci-> identification of KLK7 as protease target, serpin-activity dependence of vaspin function).

The relation of serum FABP4 levels with serum vaspin is interesting and the authors speculate on a protective role of vaspin in inflammatory processes and glucose metabolism. Yet again, the literature on mechanistic insight into vaspins anti-inflammatory roles in various tissues is not mentioned or discussed.

The study of Liu et al (2013) is not suited to demonstrate vaspins anti-diabetic and anti-obesity potential. Vaspin effects on adipogenesis are inconsistent (see e.g. Zieger et al. 2018 Mol Cell Endocrinol, and other papers were key to reveal vaspin function in adipose tissue (e.g. Hida et al. 2005 PNAS, Nakastuka et al. 2012 Diabetes).

Determination of circulating adipokine concentrations is the basis of this small pilot study. Unfortunately, information provided section on the ELISA kits used is minimal. As many ELISA kits on the market are not useful, unreliable or at least of questionable quality, it is mandatory to provide detailed information on manufacturers (not distributors), product and lot numbers, inter- and intra assay variations, controls used etc. Without this information, presented data cannot be evaluated properly. Please provide all relevant information on ELISA kits used in this study.

Author Response

           14 March 2019

Dear Reviewer,

“Medicina”

We would like to express our gratitude for your constructive and helpful comments, which have made significant contribution to the quality of our paper.

Having analyzed the comments and following your advice, we have decided to introduce certain changes into our paper. We do hope theses will bring considerable improvement to our manuscript.

Please find enclosed our responses to your comments:

“Trojnar et al. report a small pilot study that addressed vaspin concentrations in serum and urine of women with excessive gestational weight gain (EGWG). The authors found significantly lower vaspin levels in EGWG women and report interesting data that suggests a relationship of circulating vaspin and FABP4. Unfortunately, the authors fall short of discussing their data with respect to the recent insight gained on vaspin function in vitro and in vivo. Many related and highly relevant and interesting papers with respect to the authors findings are not discussed.

Instead, citation of review literature and in general papers only marginally related to the topic is excessive. The authors should focus on the relevant primary literature. This clearly decreases the enthusiasm for this manuscript and does not help the interested reader.

This is especially important, when introducing and discussing vaspin function to interpret the authors findings. Yet, findings of important papers are not discussed (e.g. Nakatsuka et al. 2012 Diabetes-> vaspin KO and tg mice, identification of GRP78 as cell surface receptor; Heiker et al. 2013 Cell Mol Life Sci-> identification of KLK7 as protease target, serpin-activity dependence of vaspin function).

The relation of serum FABP4 levels with serum vaspin is interesting and the authors speculate on a protective role of vaspin in inflammatory processes and glucose metabolism. Yet again, the literature on mechanistic insight into vaspins anti-inflammatory roles in various tissues is not mentioned or discussed.”

The whole paragraph in the Discussion section has been modified - certain information was erased and at the same time new findings on vaspin mechanism of action and its’ in vitro as well as in vivo metabolic properties have been reported, accordingly. We analyzed the manuscripts suggested by the Reviewer - these have been added to the References and renumbered, as the list has now expanded.

The paragraph in focus reads as follows:

It is still not clear why the concentration of vaspin decreases in the group of GDM women at term. There is no consensus regarding any mechanism in which vaspin alters glucose metabolism or insulin sensitivity, either. Blüher [26] proposed an interaction with a protease. In different in vitro research, human kallikrein 7 (hK7) was found to be the first target protease of vaspin, which confirmed strong inhibitory properties of vaspin. HK7 is known to be engaged in the process of insulin degradation, which seems in line with the observed increased insulin concentration of isolated pancreatic islets treated with vaspin [33]. An interesting study by Nakatsuka et al. [34] confirmed that vaspin significantly improved insulin sensitivity in vivo, in an experiment carried out on transgenic mice. The authors concluded that vaspin secreted by the adipose visceral tissue targeted hepatocytes by interacting with 78 kDa glucose-regulated protein (GRP78) molecule. In the postulated mechanism of action, the mentioned adipokine is believed to modulate the endoplasmic reticulum stress responses acting as a ligand for plasma membrane associated GRP78. The observed beneficial metabolic outcome was related to decreased gluconeogenesis and the limited synthesis of fatty acids. A detailed assessment of metabolic impact of vaspin was performed in an animal model of abdominal obesity with T2DM. The authors proved that expression of this adipokine in rat adipocytes was high when obesity, animal body weight, and insulin levels peaked in the experiment. The tissue expression of vaspin decreased as diabetes worsened, whereas  administration of vaspin to obese mice fed with high-fat high sucrose chow improved glucose tolerance and insulin sensitivity [11].

Further in the text some more modifications to the original manuscript version have been applied. In the section listed below anti-inflammatory properties of vaspin are highlighted, as requested. Please be informed that we have also added extra information regarding FGF21 – as suggested by Reviewer 2.

There is some emerging evidence to prove that vaspin possesses anti-inflammatory potential. It has been demonstrated that this adipokine suppresses cytokine-induced inflammation in 3T3-L1 adipocytes via inhibition of the nuclear factor kappa-light chain-enhancer of activated B cells (NFkB) pathway. Chronic inflammation and excessive macrophage infiltration are typical features of adipose tissue dysfunction in obesity. In the mentioned experiment vaspin was found to attenuate interleukin 1 ß (IL-1ß) induced pro-inflammatory cytokine response via inhibition of the NFkB signaling pathway in isolated adipocytes [40]. Nevertheless, these observations may be related to not yet fully understood relationships between vaspin and other adipokines, including the abovementioned FABP4. In the light of the recent study by Wang et al. [41], it seems to be very interesting to investigate associations between vaspin and fibroblast growth factor-21 (FGF-21) in future research. The cited authors demonstrated that FGF‐21 alleviates inflammation and that the anti‐inflammatory effects of FGF‐21 are mediated by the fibroblast growth factor receptor substrate 2/extracellular signal-regulated kinase 1/2 (FRS2/ERK1/2) signaling pathway in 3T3‐L1 preadipocytes.

“The study of Liu et al (2013) is not suited to demonstrate vaspins anti-diabetic and anti-obesity potential. Vaspin effects on adipogenesis are inconsistent (see e.g. Zieger et al. 2018 Mol Cell Endocrinol, and other papers were key to reveal vaspin function in adipose tissue (e.g. Hida et al. 2005 PNAS, Nakastuka et al. 2012 Diabetes).”

As a matter of fact we are not citing the study by Liu et al (2013) in our manuscript. Instead, their research from 2015 is discussed as it focuses on the association between vaspin and FABP4, which - we think - is very important from the perspective of the topic we are exploring in our paper.

The added references (the order has been modified):

·        Heiker, J.T.; Klöting, N.; Kovacs, P.; Kuettner, E.B.; Sträter, N.; Schultz, S.; Kern, M.; Stumvoll, M.; Blüher, M.; Beck-Sickinger, A.G. VaspinHYPERLINK "https://www.ncbi.nlm.nih.gov/pubmed/23370777" HYPERLINK "https://www.ncbi.nlm.nih.gov/pubmed/23370777"inhibitsHYPERLINK "https://www.ncbi.nlm.nih.gov/pubmed/23370777" HYPERLINK "https://www.ncbi.nlm.nih.gov/pubmed/23370777"kallikreinHYPERLINK "https://www.ncbi.nlm.nih.gov/pubmed/23370777" HYPERLINK "https://www.ncbi.nlm.nih.gov/pubmed/23370777"7HYPERLINK "https://www.ncbi.nlm.nih.gov/pubmed/23370777" by serpin mechanism. Cell. Mol. Life Sci. 2013, 70, 2569-2583. doi: 10.1007/s00018-013-1258-8.

·        Nakatsuka, A.; Wada, J.; Iseda, I.; Teshigawara, S.; Higashio, K.; Murakami, K.; Kanzaki, M.; Inoue, K.; Terami, T.; Katayama, A.; Hida, K.; Eguchi, J.; Horiguchi, C.S.; Ogawa, D.; Matsuki, Y.; Hiramatsu, R.; Yagita, H.; Kakuta, S.; Iwakura, Y.; Makino, H.  VaspinHYPERLINK "https://www.ncbi.nlm.nih.gov/pubmed/22837305" is an adipokine ameliorating ER stress in obesity as a ligand for cell-surface GRP78/MTJ-1 complex. Diabetes 2012, 61, 2823-2832. doi: 10.2337/db12-0232.

·        Zieger, K.; Weiner, J.; Krause, K.; Schwarz, M.; Kohn, M.; Stumvoll, M.; Blüher, M.; Heiker, J.T. VaspinHYPERLINK "https://www.ncbi.nlm.nih.gov/pubmed/28756250" suppresses cytokine-induced inflammation in 3T3-L1 adipocytes via inhibition of NFHYPERLINK "https://www.ncbi.nlm.nih.gov/pubmed/28756250"κHYPERLINK "https://www.ncbi.nlm.nih.gov/pubmed/28756250"B pathway. Mol. Cell. Endocrinol. 2018, 460, 181-188. doi: 10.1016/j.mce.2017.07.022.

·        Wang, N.; Zhao, T.T.; Li, S.M.; Sun, X.; Li, Z.C.; Li, Y.H.; Li, D.S.; Wang, W.F. Fibroblast Growth Factor 21HYPERLINK "https://www.ncbi.nlm.nih.gov/pubmed/30703283" Exerts its Anti-inflammatory Effects on Multiple Cell Types of Adipose Tissue in HYPERLINK "https://www.ncbi.nlm.nih.gov/pubmed/30703283"ObesityHYPERLINK "https://www.ncbi.nlm.nih.gov/pubmed/30703283". Obesity (Silver Spring) 2019, 27, 399-408. doi: 10.1002/oby.22376.

“Determination of circulating adipokine concentrations is the basis of this small pilot study. Unfortunately, information provided section on the ELISA kits used is minimal. As many ELISA kits on the market are not useful, unreliable or at least of questionable quality, it is mandatory to provide detailed information on manufacturers (not distributors), product and lot numbers, inter- and intra assay variations, controls used etc. Without this information, presented data cannot be evaluated properly. Please provide all relevant information on ELISA kits used in this study.”

The following experimental protocol was implemented:

ASSAYS

After centrifugation, all the collected maternal serum and urine samples were stored at −80 °C.

The following represents the information regarding Elisa kits used to determine concentration of various biological substances analyzed in our study:

-         vaspin - Human VASPIN Visceral Adipose Specific Serine Protease Inhibitor ELISA Kit (distributed by MyBioSource.com, San Diego, California, USA; cat. nr MBS2506005 ),

-         fatty acid-binding protein 4 – Human FABP4 Quantikine ELISA Kit (R&D Systems, Inc., Minneapolis, MN, USA; cat. nr DFBP40),

-         leptin – Human Leptin Quantikine ELISA Kit (R and D Systems, Inc., Minneapolis, MN, USA; cat. nr DLP00)

-         ghrelin (Human Appetite-regulating hormone ELISA Kit (Wuhan EIAab Science Co., Wuhan, China; cat. nr E2142h).

We used commercially available kits and proceeded in compliance with the manufacturer’s instructions via enzyme-linked immunosorbent assay (ELISA). The samples ran in duplicate and an averaged value of two samples was taken to analysis. The commercial tests that have been selected allow the assessment of circulating adipokines in serum and other body fluids.

Sensitivity (the minimum detectable dose; MDD) of samples was 7.8 pg/mL for human leptin ; 2,70-14,2pg/mL (mean MDD) was 6,55pg/mL for FABP4,          37,5 pg/mL for vaspin. Ghrelin detection range in used test was: 0.156-10ng/mL.

Reviewer 2 Report

Well written, interesting manuscript, employing appropriate methodology that will contribute to the literature.

Comments:

1.     Could the authors please clarify how the numbers of study participants were determined i.e. how was the study powered?

2.     Could the authors please discuss the potential of other metabolic factors such as FGF21 that may relate to their findings and conclusions?

Author Response

           14 March 2019

Dear Reviewer,

“Medicina”

We would like to express our gratitude for your constructive and helpful comments, which have made significant contribution to the quality of our paper.

Having analyzed the comments and following your advice, we have decided to introduce certain changes into our paper. We do hope theses will bring considerable improvement to our manuscript.

Please find enclosed our responses to your comments:

 “Well written, interesting manuscript, employing appropriate methodology that will contribute to the literature.

Comments:
1. Could the authors please clarify how the numbers of study participants were determined i.e. how was the study powered?”

We have performed pre-study calculation of the required sample size and determined the number of participants required in each group (EGWG vs control) for 25 females. Furthermore, the retrospective analysis revealed that the power of the test assuming an effect size equals to the effect size observed in the current sample.

“2. Could the authors please discuss the potential of other metabolic factors such as FGF21 that may relate to their findings and conclusions?”

The following section was added to the Discussion, in which the role of FGF-21 is highlighted with reference to vaspin, the adipokine we are focusing on in the manuscript. The modified version now reads as follows:

Nevertheless, these observations may be related to not yet fully understood relationships between vaspin and other adipokines, including the abovementioned FABP4. In the light of the recent study by Wang et al. [41], it seems to be very interesting to investigate associations between vaspin and fibroblast growth factor-21 (FGF-21) in future research. The cited authors demonstrated that FGF‐21 alleviates inflammation and that the anti‐inflammatory effects of FGF‐21 are mediated by the fibroblast growth factor receptor substrate 2/extracellular signal-regulated kinase 1/2 (FRS2/ERK1/2) signaling pathway in 3T3‐L1 preadipocytes.
